# Separating parental and treatment contributions to perinatal health after fresh and frozen embryo transfer in assisted reproduction: A cohort study with within-sibship analysis

Kjersti Westvik-Johari[1,2]*, Liv Bente Romundstad[3,4], Deborah A. Lawlor[5,6,7], Christina Bergh[8], Mika Gissler[9,10], Anna-Karina A. Henningsen[11], Siri E. Håberg[3], Ulla-Britt Wennerholm[8], Aila Tiitinen[12], Anja Pinborg[11], Signe Opdahl[2]

1 Department of Fertility, Women and Children's Centre, St. Olavs Hospital, Trondheim, Norway, 2 Department of Public Health and Nursing, Norwegian University of Science and Technology, Trondheim, Norway, 3 Centre for Fertility and Health, Norwegian Institute of Public Health, Oslo, Norway, 4 Spiren Fertility Clinic, Trondheim, Norway, 5 MRC Integrative Epidemiology Unit, University of Bristol, Bristol, United Kingdom, 6 Population Health Sciences, Bristol Medical School, Bristol, United Kingdom, 7 NIHR Bristol Biomedical Research Centre, Bristol, United Kingdom, 8 Department of Obstetrics and Gynaecology, Institute of Clinical Sciences, Sahlgrenska Academy, University of Gothenburg, Sahlgrenska University Hospital, Gothenburg, Sweden, 9 Statistics and Registers Unit, Finnish Institute for Health and Welfare, Helsinki, Finland, 10 Department of Neurobiology, Care Sciences and Society, Karolinska Institutet, Stockholm, Sweden, 11 Fertility Clinic, Rigshospitalet, Copenhagen University Hospital, Copenhagen, Denmark, 12 Department of Obstetrics and Gynaecology, University of Helsinki and Helsinki University Hospital, Helsinki, Finland

* kjersti.westvik-johari@ntnu.no

**Data Availability Statement:** Data cannot be shared publicly owing to restrictions by law. Data

## Abstract

### Background

Compared to naturally conceived children, adverse perinatal outcomes are more common among children born after assisted reproductive technology with fresh embryo transfer (fresh-ET) or frozen embryo transfer (frozen-ET). However, most previous studies could not adequately control for family confounding factors such as subfertility. We compared birth size and duration of pregnancy among infants born after fresh-ET or frozen-ET versus natural conception, using a within-sibship design to account for confounding by maternal factors.

### Methods and findings

This registry-based cohort study with nationwide data from Denmark (1994–2014), Norway (1988–2015), and Sweden (1988–2015) consisted of 4,510,790 live-born singletons, 4,414,703 from natural conception, 78,095 from fresh-ET, and 17,990 from frozen-ET. We identified 33,056 offspring sibling groups with the same mother, conceived by at least 2 different conception methods. Outcomes were mean birthweight, small and large for gestational age, mean gestational age, preterm (<37 weeks, versus ≥37), and very preterm birth (<32 weeks, versus ≥32). Singletons born after fresh-ET had lower mean birthweight (−51 g, 95% CI −58 to −45, $p < 0.001$) and increased odds of small for gestational age (odds

are available from the CoNARTaS server at Statistics Denmark, after approval by the Ethics Committees and registry keeping authorities in each country, as described in the following publication: Opdahl S, Henningsen AA, Bergh C, Gissler M, Romundstad LB, Petzold M, Tiitinen A, Wennerholm UB, Pinborg AB. Data Resource Profile: Committee of Nordic Assisted Reproductive Technology and Safety (CoNARTaS) cohort. Int J Epidemiol. 2020 Apr 1;49(2):365-366f. doi: 10.1093/ije/dyz228. Contact information for Statistics Denmark: Division of Research Services Statistics Denmark Sejrøgade 11 DK-2100 Copenhagen Denmark E-mail: forskningsservice@dst.dk Phone: +45 39 17 31 30.

**Funding:** This work was supported by the NordForsk (grant number 71450, AP), the Central Norway Regional Health Authorities (grant number 46045000, LBR), the Nordic Federation of Obstetrics and Gynaecology (grant numbers NF13041, NF15058, NF16026 and NF17043, UBW & AT), the Interreg Øresund-Kattegat-Skagerrak European Regional Development Fund (ReproUnion project, AP & CB), and by the Research Council of Norway's Centre of Excellence funding scheme (grant number 262700, SEH & LBR). DAL's contribution to this work is supported by the University of Bristol and Medical Research Council (MC_UU_00011/6, DAL), the Bristol National Institute of Health (NIHR) Research Biomedical Research Centre (DAL) and a NIHR Senior Investigator award (NF-0616-10102, DAL). The funders had no role in study design, data collection and analysis, decision to publish, or preparation of the manuscript.

**Competing interests:** The authors have declared that no competing interests exist.

**Abbreviations:** ART, assisted reproductive technology; CoNARTaS, Committee of Nordic Assisted Reproductive Technology and Safety; fresh-ET, fresh embryo transfer; frozen-ET, frozen embryo transfer; OR, odds ratio.

ratio [OR] 1.20, 95% CI 1.08 to 1.34, $p < 0.001$), while those born after frozen-ET had higher mean birthweight (82 g, 95% CI 70 to 94, $p < 0.001$) and increased odds of large for gestational age (OR 1.84, 95% CI 1.56 to 2.17, $p < 0.001$), compared to naturally conceived siblings. Conventional population analyses gave similar results. Compared to naturally conceived siblings, mean gestational age was lower after fresh-ET (−1.0 days, 95% CI −1.2 to −0.8, $p < 0.001$), but not after frozen-ET (0.3 days, 95% CI 0.0 to 0.6, $p = 0.028$). There were increased odds of preterm birth after fresh-ET (OR 1.27, 95% CI 1.17 to 1.37, $p < 0.001$), and in most models after frozen-ET, versus naturally conceived siblings, with somewhat stronger associations in population analyses. For very preterm birth, population analyses showed increased odds for both fresh-ET (OR 2.03, 95% CI 1.90 to 2.12, $p < 0.001$) and frozen-ET (OR 1.66, 95% CI 1.42 to 1.94, $p < 0.001$) compared with natural conception, but results were notably attenuated within siblings (OR 1.18, 95% CI 1.0 to 1.41, $p = 0.059$, and OR 0.92, 95% CI 0.67 to 1.27, $p = 0.6$, for fresh-ET and frozen-ET, respectively). Sensitivity analyses in full siblings, in siblings born within 3-year interval, by birth order, and restricting to single embryo transfers and blastocyst transfers were consistent with the main analyses. Main limitations were high proportions of missing data on maternal body mass index and smoking.

## Conclusions

We found that infants conceived by fresh-ET had lower birthweight and increased odds of small for gestational age, and those conceived by frozen-ET had higher birthweight and increased odds of large for gestational age. Conception by either fresh-ET or frozen-ET was associated with increased odds of preterm birth. That these findings were observed within siblings, as well as in conventional multivariable population analyses, reduces the likelihood that they are explained by confounding or selection bias.

## Trial registration

ClinicalTrials.gov ISRCTN11780826.

## Author summary

### Why was this study done?

- Children born after assisted reproductive technology have more adverse perinatal outcomes than naturally conceived children, which differ according to treatment method.

- It is unknown to what extent these associations result from the fertility treatment or from confounding by underlying maternal or family factors.

### What did the researchers do and find?

- Using health registry data from Denmark, Norway, and Sweden, we compared perinatal health after fresh embryo transfer (fresh-ET) or frozen embryo transfer (frozen-ET) to that after natural conception, in a cohort of 4,606,875 newborns. In addition, we

compared siblings conceived by different methods to account for family confounding (*n* = 33,056 sibling groups).

- We found that children conceived by frozen-ET have a higher birthweight and higher risk of large for gestational age, whereas children conceived by fresh-ET have a lower birthweight and higher risk of small for gestational, compared to naturally conceived children, both in the population and within siblings.

- Within sibships, children conceived by fresh-ET and frozen-ET had increased risks of preterm birth (<37 weeks) of similar magnitude, while neither fresh-ET nor frozen-ET was associated with risk of very preterm birth (<32 weeks), despite strong associations for both outcomes in population analyses.

## What do these findings mean?

- Fresh-ET and frozen-ET showed opposite associations with birthweight, but similar associations with preterm birth, after controlling for measured and unmeasured family-level confounding.

- Both treatments are associated with adverse perinatal outcomes, in comparison to natural conception. Our findings provide important information that can be used by couples and their clinicians in making decisions about which type of ART to undertake.

## Introduction

The use of assisted reproductive technology (ART) is increasing worldwide, and children born after ART now comprise more than 7% of births in some countries [1–4]. The number of children born after fresh embryo transfer (fresh-ET) has increased steadily over 3 decades, and the number of children born after frozen embryo transfer (frozen-ET) has increased sharply during the last decade [1,2,5]. Whilst elective single embryo transfer has reduced multiple pregnancy and adverse outcomes associated with that [5,6], singleton ART newborns still have worse perinatal outcomes compared with naturally conceived newborns [7]. Meta-analyses show lower birthweight, lower gestational age, higher risk of small for gestational age, and higher risk of preterm birth among newborns after fresh-ET compared to naturally conceived newborns [8,9]. In contrast, newborns after frozen-ET have lower risk of small for gestational age and preterm birth compared to newborns after fresh-ET [10,11], but higher mean birthweight and higher risk of large for gestational age compared to naturally conceived newborns [8,12–14]. Most previous studies have not adequately controlled for family confounding factors, such as maternal health and socioeconomic position [8,14,15]. Subfertile couples who conceive while awaiting ART treatment have suboptimal perinatal outcomes compared to fertile couples, indicating that parental factors contribute to the adverse events [3]. Without attempts to control for potential family confounding, it is unclear whether these associations are attributable to treatment.

Comparing siblings born after different conception methods offers an alternative approach to conventional multivariable analyses in unrelated children, and may help disentangle the contributions from ART treatment, shared genetics, parental health factors, and confounding

from, for example, background family socioeconomic position [16,17]. Four previous studies with a sibling design compared any ART conception with natural conception, and all reported lower birthweight and shorter gestational duration in infants conceived by ART, though for some outcomes, wide confidence intervals included the null [18–21]. A Danish study could differentiate between fresh and frozen transfer and found lower birthweight and shorter gestation for fresh-ET compared to naturally conceived siblings (3,879 sibling pairs) and higher birthweight for frozen-ET compared to fresh-ET siblings (358 sibling pairs) [22,23]. An American study included only children conceived after ART and found that children conceived by frozen-ET had higher birthweights and higher risk of large for gestational age than their fresh-ET siblings (3,681 pairs) [22]. None of the previous studies compared children conceived by frozen-ET to naturally conceived siblings, which is a necessary comparison to understand whether the higher birthweights and increased risk of large for gestational age associated with frozen-ET simply reflect the observed lower birthweight for fresh-ET compared with natural conception.

The aim of this study was to determine the associations of fresh-ET and frozen-ET, compared to natural conception, with birth size and duration of pregnancy. We used nationwide data from 3 countries that provided a sufficiently large sample size to precisely estimate associations using a within-sibship design. The within-sibship analysis assumes that most confounders are at the family level and that there is very little individual-level confounding. Specifically, in this study we assume that in the within-sibship analyses we can control for unmeasured confounding by shared family factors, such as socioeconomic position, underlying maternal health, and health behaviors [16,17,24].

## Methods

### Data sources

This cohort study is based on the Committee of Nordic Assisted Reproductive Technology and Safety (CoNARTaS) cohort [5], which includes data on all births registered in the nationwide medical birth registries in Denmark (1994–2014), Norway (1984–2015), and Sweden (1985–2015). Children born after ART were identified through data linkage with the national ART registries and databases, using the unique national identity number assigned to each resident. The registration of ART pregnancies was initiated at different times in each country. In Denmark, all ART cycles from both public and private clinics have been registered in the national ART registry since 1994, resulting in almost 100% completeness [25]. Since 1984, Norwegian public and private ART clinics send notifications to the Medical Birth Registry for all ART cycles that result in pregnancy verified by ultrasound in gestational week 6–7. In Sweden, deliveries after ART were reported to the National Board of Health and Welfare from 1982 to 2006. Since 2007, all ART cycles in Sweden are reported to the National Quality Registry for Assisted Reproduction.

### Exposures, outcomes, and covariates

Exposures were ART conception with fresh-ET or frozen-ET versus natural conception (the reference group). Fresh-ET and frozen-ET were defined based on treatment entries in the ART registries/databases. Frozen-ET included both first embryo transfer (i.e., when a "freeze-all" treatment was undertaken) as well as those with a subsequent transfer after an initial fresh transfer. Natural conceptions were defined based on any registered pregnancy with no registration of ART conception.

We defined perinatal health outcomes as birth size (birthweight, small for gestational age, and large for gestational age) and duration of pregnancy (gestational age at birth, preterm

birth, and very preterm birth). Birthweight was measured in grams. We used Marsal's equations for intrauterine growth to estimate $z$-scores of birthweights, where 1 standard deviation was set to 11% of the expected birthweight according to sex and gestational age [26]. Small for gestational age was defined as birthweight $< -2$ standard deviations, and large for gestational age was defined as birthweight $> +2$ standard deviations from expected mean birthweight. For natural conceptions, gestational age was reported in days and estimated by routine ultrasound examination, performed in week 18–20 of pregnancy in Norway and Sweden, and in late first trimester in Denmark. If this information was missing, the date of last menstrual period was used to calculate gestational age. For ART pregnancies, gestational age was estimated based on embryo transfer in Sweden, while in Norway and Denmark the first trimester (Denmark) or week 18–20 (Norway) ultrasound screening was used, and only if this was missing was the date of embryo transfer used. Preterm birth was defined as birth before 37 weeks of gestation, versus at $\geq 37$ weeks, and very preterm birth as birth before 32 weeks of gestation. Maternal and paternal identity codes were recorded in the medical birth registries, with paternal identity available for 98% of newborns. For our main analyses, we identified siblings as children with the same mother from the maternal identity code. In sensitivity analyses, we repeated analyses using full siblings (same mother and father) identified using the maternal and paternal identity codes.

Potential confounders were defined as any factor that could plausibly influence the need for ART, birthweight, or gestational age; these were identified based on previous literature. We adjusted for the following observed confounders: country, year of birth, and maternal age, parity, BMI, height, and smoking. Maternal BMI was calculated as weight in kilograms divided by height in square meters, based on pre-pregnancy or first trimester values and categorized as underweight ($<18.5$ kg/m$^2$), normal (18.5–24.9 kg/m$^2$), overweight (25.0–29.9 kg/m$^2$), and obese ($\geq 30.0$ kg/m$^2$). Further, we categorized maternal height ($<150$, 150–159, 160–169, 170–179, or $\geq 180$ cm), smoking (yes or no, where yes was any smoking during pregnancy), and parity (number of previous deliveries: 0, 1, 2, or 3). Maternal age and offspring year of birth were used as continuous variables. Smoking was registered throughout the study period in Denmark and Sweden and since 1999 in Norway. Maternal height and weight were registered in 1988–1989 and 1992–2015 in Sweden, with substantial missing data in the early years. In Denmark and Norway, registration of maternal height and weight was implemented from 2004 and 2007, respectively, also with substantial missing data during the first years of registration. In addition to these observed confounders, we considered parental socioeconomic position to be a key confounder, but we did not have data on individual income or education. However, the sibship analysis approach controls for this family-level confounding on the assumption that parental socioeconomic position is likely to be very similar between siblings.

## Study population

Fig 1 shows the flow of participants into the main analysis and sensitivity analysis datasets. We defined our study period as being from 1988, when the first child born after embryo cryopreservation was registered in our data (from 1994 for Denmark, as data were not available until then), until 2014 (Denmark) and 2015 (Norway and Sweden). Eligibility was defined as liveborn singletons whose mothers delivered their first child within the study period and were age 20 years or older at their first delivery (4,617,121 infants with 2,390,386 mothers). These criteria ensured comparability of maternal age between ART and natural conceptions while maximizing the number of sibling groups in the analysis sample. We excluded all singletons with unknown parity in pregnancies after the first birth, maternal age $\geq 45$ years, and parity $\geq 4$, as there were very few ART births to mothers with 4 or more deliveries. We further excluded

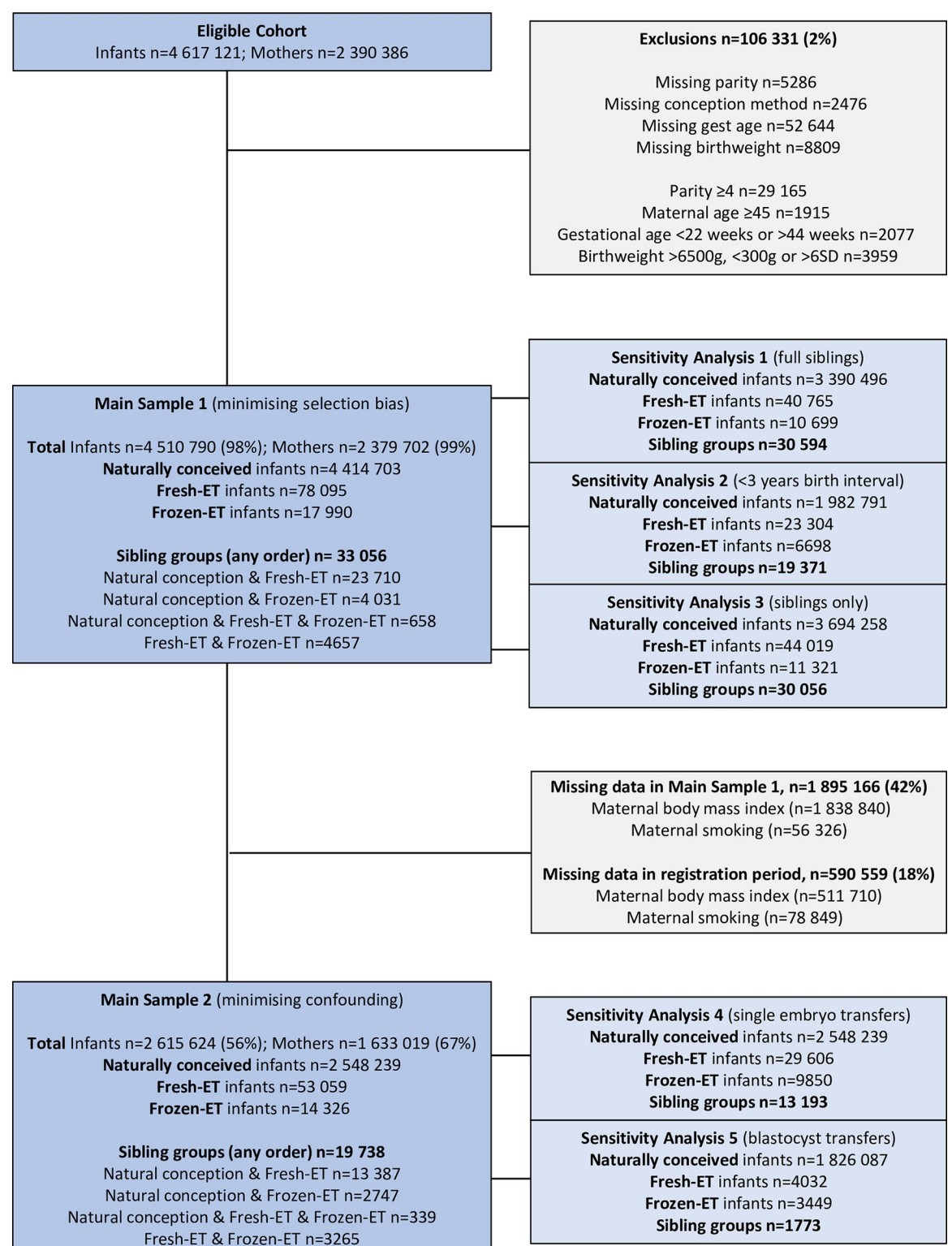

**Fig 1. Flow chart of the study population.** If not otherwise specified, sibling groups refer to maternal offspring siblings conceived through at least 2 of the 3 different conception methods. Fresh-ET, fresh embryo transfer; Frozen-ET, frozen embryo transfer; gest, gestational.

singletons with unknown conception method, gestational age, or birthweight, as well as singletons with extreme values of gestational age ($<22$ weeks or $>44$ weeks) and birthweight ($>6,500$ g, $<300$ g, or $>6$ SD). After these exclusions, our main sample 1 (largest sample, with lowest risk of selection bias) comprised 4,510,790 infants with 2,379,702 mothers, where 78,095 were born from fresh-ET and 17,990 were born from frozen-ET. In this sample there were 33,056 sibling groups with at least 2 of the 3 different conception methods, including 24,368 sibling groups with both fresh-ET and natural conception and 4,689 sibling groups with both frozen-ET and natural conception. For main sample 2 (smallest sample, with maximum confounder adjustment), we restricted main sample 1 to deliveries with complete data on maternal BMI and smoking (58% of main sample 1). Corresponding numbers for main sample 2 were 2,615,624 infants with 1,633,019 mothers, including 53,059 born after fresh-ET and 14,326 born after frozen-ET. In main sample 2, there were 20,227 sibling groups with at least 2 of the 3 conception methods, including 13,869 sibling groups with both fresh-ET and natural conception and 3,168 sibling groups with both frozen-ET and natural conception. To explore whether the results were driven by specific subgroups or whether the associations were influenced by which conception method occurred first, we identified each mother's 2 first consecutive deliveries and categorized them by order of conception method. In main sample 2, this gave a total of 698,990 offspring sibling groups that belonged to 1 of 9 possible sibling combinations.

## Statistical analysis

We used multilevel linear and logistic models to compare outcomes across conception methods with children as one level and mothers as another. We used random effects models for conventional population estimates and fixed effects models for sibship comparisons (i.e., comparisons within sibships). Precision was estimated by 95% confidence intervals (CIs). To increase interpretability of the odds ratios (ORs), we used post-estimation commands to obtain absolute risks and risk differences. The within-sibling estimates were based on siblings who were discordant for conception method. Population estimates in main sample 1 were adjusted for year of birth, country, maternal age, and parity. In main sample 2 we additionally adjusted for height, pre-pregnancy or first trimester BMI, and smoking status during pregnancy. The sibling comparisons were adjusted for the same covariates except country and height, which are stable within mothers.

We performed several sensitivity analyses to test the robustness of our findings (Fig 1). First, we explored the importance of constant paternal factors by repeating analyses on full siblings only (same mother and father). Second, we restricted analyses to siblings born within a 3-year interval as both family background and health are likely to be more constant among siblings born within a short timeframe. Third, we restricted the population-level analyses to siblings (excluding all infants where the mother had only 1 child in the sample). This enabled us to explore whether any differences between population and sibling results might be driven by families with only 1 child being different to those with 2 or more. Finally, we restricted the ART population to single embryo transfers and to blastocyst transfers (i.e., culture duration 5–6 days) to account for changes in practice over the study period that could potentially influence our results [5,27,28]. Single, compared with double, embryo transfer is associated with higher [29] and eliminates vanishing twin syndrome, which may also influence birthweight and gestational duration [30]. Blastocysts are exposed to culture medium and other in vitro conditions for a longer period than cleavage stage embryos (5–6 versus 2–3 days), and this may also influence fetal growth and gestational duration.

This study is reported as per the Strengthening the Reporting of Observational Studies in Epidemiology (STROBE) guideline (S1 STROBE Checklist). Our analyses were planned in

advance of the research team accessing any data, and our study protocol is provided (S1 Study Protocol). The CoNARTaS project is also registered in the ISRCTN registry (ISRCTN11780826).

### Ethical approval

In Denmark, ethical approval is not required for scientific projects based solely on registry data. In Norway, ethical approval was given by the Regional Committee for Medical and Health Research Ethics (REK-Nord, 2010/1909). In Sweden approval was obtained from the ethical committee in Gothenburg (Dnr214-12, T422-12, T516-15, T233-16, T300-17, T1144-17, and T121-18).

### Patient and public involvement

This study was a secondary data analysis and was done without patient involvement. Patients were not involved in setting the research question or the outcome measures, nor were they involved in developing plans for the design or implementation of the study.

## Results

### Baseline characteristics

Mothers who conceived by ART were older and more commonly primiparous than mothers of naturally conceived children (Table 1). While the naturally conceived children were evenly distributed through the study period, more than 80% of ART children were born after 2002. Mean maternal height and BMI were comparable between all conception groups. Fewer ART mothers were underweight and obese, but there was a higher proportion in the overweight category. Among children born after fresh-ET, about 42% of the children were conceived by intracytoplasmic sperm injection, 47.4% were born after single embryo transfer, and only 5.7% were born after blastocyst transfer. Similar proportions were found in children born after frozen-ET, with 40.2% conceived by intracytoplasmic sperm injection, 64.4% born after single embryo transfer, though 20.8% born after blastocyst transfer.

### Birthweight

In population analyses in both samples, children from fresh-ET were on average lighter, and those born after frozen-ET were on average heavier, at birth compared to naturally conceived children, after adjustment for all observed confounders (Tables 2 and 3). Analyses of birthweight $z$-score according to gestational age and sex showed similar patterns. Children born after fresh-ET had a higher risk of small for gestational age and lower risk of large for gestational age compared to naturally conceived children, whereas the opposite was true for children conceived via frozen-ET (Fig 2A; Tables A and B in S1 Text). The sibship comparisons showed the same patterns, with clear differences in mean birthweight and small and large for gestational age between children born after fresh-ET and frozen-ET, compared to their siblings who were naturally conceived. The magnitudes of association were also similar between sibship and population-level analyses (Fig 2A).

### Duration of pregnancy

In both main samples, gestational age was shorter for children of fresh-ET and frozen-ET in population analyses (Tables 2 and 3). In the corresponding sibling comparisons, children of fresh-ET had gestational ages closer to their naturally conceived siblings, while children of frozen-ET had a longer gestational age compared to their naturally conceived siblings. In

**Table 1. Characteristics of the 4,510,790 live-born singletons in main sample 1.**

| Characteristic | Natural conception | | Fresh embryo transfer | | Frozen embryo transfer | |
|---|---|---|---|---|---|---|
| | *n* or mean | Percent or SD | *n* or mean | Percent or SD | *n* or mean | Percent or SD |
| Participants (*n*, %) | 4,414,703 | 97.9 | 78,095 | 1.7 | 17,990 | 0.4 |
| Country (*n*, %) | | | | | | |
| Denmark | 977,754 | 22.2 | 25,041 | 32.1 | 3,347 | 18.6 |
| Norway | 1,193,617 | 27.0 | 16,551 | 21.2 | 3,283 | 18.3 |
| Sweden | 2,243,334 | 50.8 | 36,503 | 46.7 | 11,360 | 63.2 |
| Birth year (*n*, %) | | | | | | |
| 1988–1996 | 1,020,394 | 23.1 | 5,762 | 7.4 | 494 | 2.8 |
| 1997–2001 | 806,469 | 18.3 | 11,190 | 14.3 | 1,098 | 6.1 |
| 2002–2006 | 909,995 | 20.6 | 17,727 | 22.7 | 2,541 | 14.1 |
| 2007–2011 | 965,027 | 21.9 | 24,346 | 31.2 | 6,499 | 36.1 |
| 2012–2015 | 712,820 | 16.2 | 19,070 | 24.4 | 7,358 | 40.1 |
| Parity (*n*, %) | | | | | | |
| 0 | 2,258,213 | 51.2 | 58,739 | 75.2 | 10,413 | 57.9 |
| 1 | 1,585,604 | 35.9 | 16,977 | 21.7 | 6,539 | 36.5 |
| 2 | 476,823 | 10.8 | 2,039 | 2.6 | 920 | 5.1 |
| 3 | 94,065 | 2.1 | 340 | 0.4 | 118 | 0.7 |
| Maternal age, in years (mean, SD) | 29.6 | 4.8 | 33.8 | 4.2 | 34.3 | 4.1 |
| Sex (*n*, %) | | | | | | |
| Boys | 2,269,179 | 51.4 | 39,914 | 51.1 | 9,200 | 51.1 |
| Girls | 2,145,526 | 48.6 | 38,181 | 48.9 | 8,790 | 48.9 |
| Smoking | | | | | | |
| Yes | 447,967 | 10.2 | 4,040 | 5.2 | 540 | 3.0 |
| Missing (%) | | 15.4 | | 9.4 | | 6.1 |
| Maternal height, in cm (mean, SD) | 166.8 | 6.3 | 167.7 | 6.4 | 167.5 | 6.5 |
| Missing (%) | | 35.5 | | 27.9 | | 2.8 |
| Maternal BMI, in kg/m$^2$ (mean, SD) | 24.2 | 4.5 | 24.2 | 4.1 | 24.2 | 4.0 |
| Missing (%) | | 41.0 | | 30.8 | | 19.3 |
| Maternal BMI, in kg/m$^2$ (*n*, %) | | | | | | |
| <18.5 | 80,471 | 3.2 | 1,256 | 2.4 | 310 | 2.2 |
| 18.5–24.9 | 1,627,235 | 63.9 | 33,733 | 63.6 | 9,096 | 63.5 |
| 25.0–29.9 | 575,551 | 22.6 | 12,938 | 28.8 | 3,541 | 24.7 |
| ≥30.0 | 264,982 | 10.4 | 5,133 | 9.7 | 1,379 | 9.6 |
| Fertilization method (*n*, %) | | | | | | |
| IVF | — | — | 44,474 | 58.0 | 9,818 | 59.8 |
| ICSI | — | — | 32,164 | 42.0 | 6,597 | 40.2 |
| Embryos transferred (*n*, %) | | | | | | |
| 1 | — | — | 36,992 | 47.4 | 11,577 | 64.4 |
| 2 | — | — | 29,915 | 38.3 | 4,197 | 23.3 |
| 3 | — | — | 1,880 | 2.4 | 128 | 0.7 |
| Unknown | | | 9,308 | 11.9 | 2,088 | 11.6 |
| Embryo culture duration, in days (*n*, %) | | | | | | |
| 2–3 | — | — | 61,654 | 79.0 | 11,695 | 65.0 |
| 5–6 | — | — | 4,437 | 5.7 | 3,748 | 20.8 |
| Unknown | | | 12,004 | 15.4 | 2,547 | 14.2 |

BMI, body mass index; ICSI, intracytoplasmic sperm injection; IVF, in vitro fertilization; SD, standard deviation.

**Table 2. Birthweight and gestational age by conception method: Population estimates and within-sibship estimates in main sample 1 (minimizing selection).**

| Outcome and conception method | Population estimates (random effects) | | | | | Within-sibship estimates (fixed effects) | | | | |
|---|---|---|---|---|---|---|---|---|---|---|
| | Number | Mean[1] | Mean difference[1] | Adj. mean difference[2] | 95% CI | Number[3] | Mean[1] | Mean difference[1] | Adj. mean difference[2] | 95% CI |
| **Birthweight, grams** | | | | | | | | | | |
| Natural conception | 4,414,703 | 3,541 | 0 | 0 | Ref. | 33,889 | 3,540 | 0 | 0 | Ref. |
| Fresh-ET | 78,095 | 3,410 | −127 | −71 | −75 to −67 | 30,167 | 3,424 | −116.3 | −51 | −58 to −45 |
| Frozen-ET | 17,990 | 3,581 | 51 | 66 | 59 to 74 | 9,589 | 3,623 | 83 | 82 | 70 to 94 |
| **Birthweight, z-score** | | | | | | | | | | |
| Natural conception | 4,414,703 | −0.01 | 0 | 0 | Ref. | 33,889 | −0.01 | 0 | 0 | Ref. |
| Fresh-ET | 78,095 | −0.19 | −0.18 | −0.05 | −0.06 to −0.04 | 30,167 | −0.23 | −0.22 | −0.06 | −0.78 to −0.05 |
| Frozen-ET | 17,990 | 0.16 | 0.18 | 0.21 | 0.18 to 0.21 | 9,589 | 0.2 | 0.21 | 0.19 | 0.17 to 0.22 |
| **Gestational age, days** | | | | | | | | | | |
| Natural conception | 4,414,703 | 279.1 | 0 | 0 | Ref. | 33,889 | 279.0 | 0 | 0 | Ref. |
| Fresh-ET | 78,095 | 276.6 | −2.3 | −2.1 | −2.2 to −2.0 | 30,167 | 277.9 | −1.1 | −1.0 | −1.2 to −0.8 |
| Frozen-ET | 17,990 | 278.1 | −0.8 | −0.6 | −0.8 to −0.4 | 9,589 | 279.2 | 0.2 | 0.3 | 0.0 to 0.6 |

Adj., adjusted; CI, confidence interval; fresh-ET, fresh embryo transfer; frozen-ET, frozen embryo transfer; Ref., reference.

[1]Unadjusted.

[2]Adjusted for maternal age, parity, and year of birth. Random effects are additionally adjusted for country.

[3]Number of children that are part of a sibling group with at least 2 different conception methods within the group.

population analysis, in both main sample 1 and main sample 2, children conceived with fresh-ET and frozen-ET had substantially higher odds of preterm and very preterm birth than naturally conceived children. For preterm birth, there was some attenuation in sibling compared to population analyses (Fig 2; Tables A and B in S1 Text), particularly for frozen-ET in main sample 1. In sibship analyses of very preterm birth, there was more marked attenuation than seen in analyses of preterm birth, with point estimates close to the null value, though with wide confidence intervals.

Fig 3D shows the risk of preterm birth according to birth order for the different combinations of conception methods in consecutive sibling pairs. All sibling groups with 1 or 2 children of ART were at higher risk compared to the naturally conceived sibling pairs, and there were no clear differences between treatment types (fresh-ET and frozen-ET) in risk of preterm birth.

## Sensitivity analyses

For the birthweight outcomes (mean birthweight, small for gestational age, and large for gestational age), the results of all sensitivity analyses (Tables C–L in S1 Text) were consistent with the findings from the main analysis. Concerning duration of pregnancy (mean gestational age, preterm birth, and very preterm birth), the sensitivity analyses were overall in line with the findings from main sample 2. One exception was the sibship comparison where ART treatment was restricted to blastocyst transfers, which may indicate an increased risk of both preterm and very preterm birth among children born after fresh-ET frozen-ET compared to their naturally conceived siblings (Table L in S1 Text). However, these estimates were imprecise due to small sample sizes.

**Table 3. Birthweight and gestational age by conception method: Population estimates and within-sibship estimates in main sample 2 (minimizing confounding).**

| Outcome and conception method | Population estimates (random effects) | | | | | Within-sibship estimates (fixed effects) | | | | |
|---|---|---|---|---|---|---|---|---|---|---|
| | Number | Mean[1] | Mean difference[1] | Adj. mean difference[2] | 95% CI | Number[3] | Mean[1] | Mean difference[1] | Adj. mean difference[2] | 95% CI |
| **Birthweight, grams** | | | | | | | | | | |
| Natural conception | 2,548,239 | 3,547 | 0 | 0 | Ref. | 19,656 | 3,547 | 0 | 0 | Ref. |
| Fresh-ET | 53,059 | 3,413 | −134 | −83 | −87 to −78 | 17,631 | 3,415 | −132 | −52 | −61 to −44 |
| Frozen-ET | 14,326 | 3,583 | 42 | 56 | 48 to 65 | 6,538 | 3,610 | 63 | 75 | 61 to 89 |
| **Birthweight, *z*-score** | | | | | | | | | | |
| Natural conception | 2,548,239 | 0.01 | 0 | 0 | Ref. | 19,656 | 0.01 | 0 | 0 | Ref. |
| Fresh-ET | 53,059 | −0.20 | −0.22 | −0.08 | −0.09 to −0.07 | 17,631 | −0.28 | −0.28 | −0.08 | −0.10 to −0.06 |
| Frozen-ET | 14,326 | 0.14 | 0.15 | 0.19 | 0.18 to 0.21 | 6,538 | 0.15 | 0.14 | 0.17 | 0.15 to 0.20 |
| **Gestational age, days** | | | | | | | | | | |
| Natural conception | 2,548,239 | 279.0 | 0 | 0 | Ref. | 19,656 | 279.0 | 0 | 0 | Ref. |
| Fresh-ET | 53,059 | 276.9 | −2.0 | −2.1 | −2.2 to −2.0 | 17,631 | 278.2 | −0.8 | −0.8 | −1.0 to −0.6 |
| Frozen-ET | 14,326 | 278.4 | −0.5 | −0.7 | −0.9 to −0.5 | 6,538 | 279.3 | 0.4 | 0.4 | −0.0 to 0.7 |

Adj., adjusted; CI, confidence interval; fresh-ET, fresh embryo transfer; frozen-ET, frozen embryo transfer; Ref., reference.

[1]Unadjusted.

[2]Adjusted for maternal age, parity, year of birth, maternal pre-pregnancy or first trimester body mass index, and maternal smoking during pregnancy. Random effects are additionally adjusted for country and maternal height.

[3]Number of children that are part of a sibling group with at least 2 different conception methods within the group.

## Discussion

### Summary of findings

We found evidence that children born after ART were at higher risk of adverse perinatal outcomes compared to the background population. Given the consistency of findings across conventional population and sibship analyses, in 2 samples (one minimizing selection bias and the other minimizing confounding) and multiple sensitivity analyses, our findings indicate that conception through fresh-ET was associated with lower mean birthweight and a higher risk of small for gestational age, whereas conception with frozen-ET was associated with higher mean birthweight and a higher risk of large for gestational age, compared to natural conception. Further, fresh-ET was associated with a shorter mean gestational age, and both fresh-ET and frozen-ET were associated with higher odds of preterm birth. Whilst population analyses suggested increased odds of very preterm birth in children conceived by either fresh-ET or frozen-ET, this was markedly attenuated in sibship analyses, though statistical power was limited in these analyses and confidence intervals were wide. The stronger associations at the population level for mean duration, preterm birth, and very preterm birth suggest that unmeasured maternal factors contribute to gestational duration in addition to the contribution of conception by either fresh-ET or frozen-ET.

### Strengths and limitations

Our study involved 2 main samples, both with detailed maternal data, including information on previous deliveries and conception method. While main sample 1 was less prone to selection bias because it consisted of an unselected and larger population, main sample 2 provided

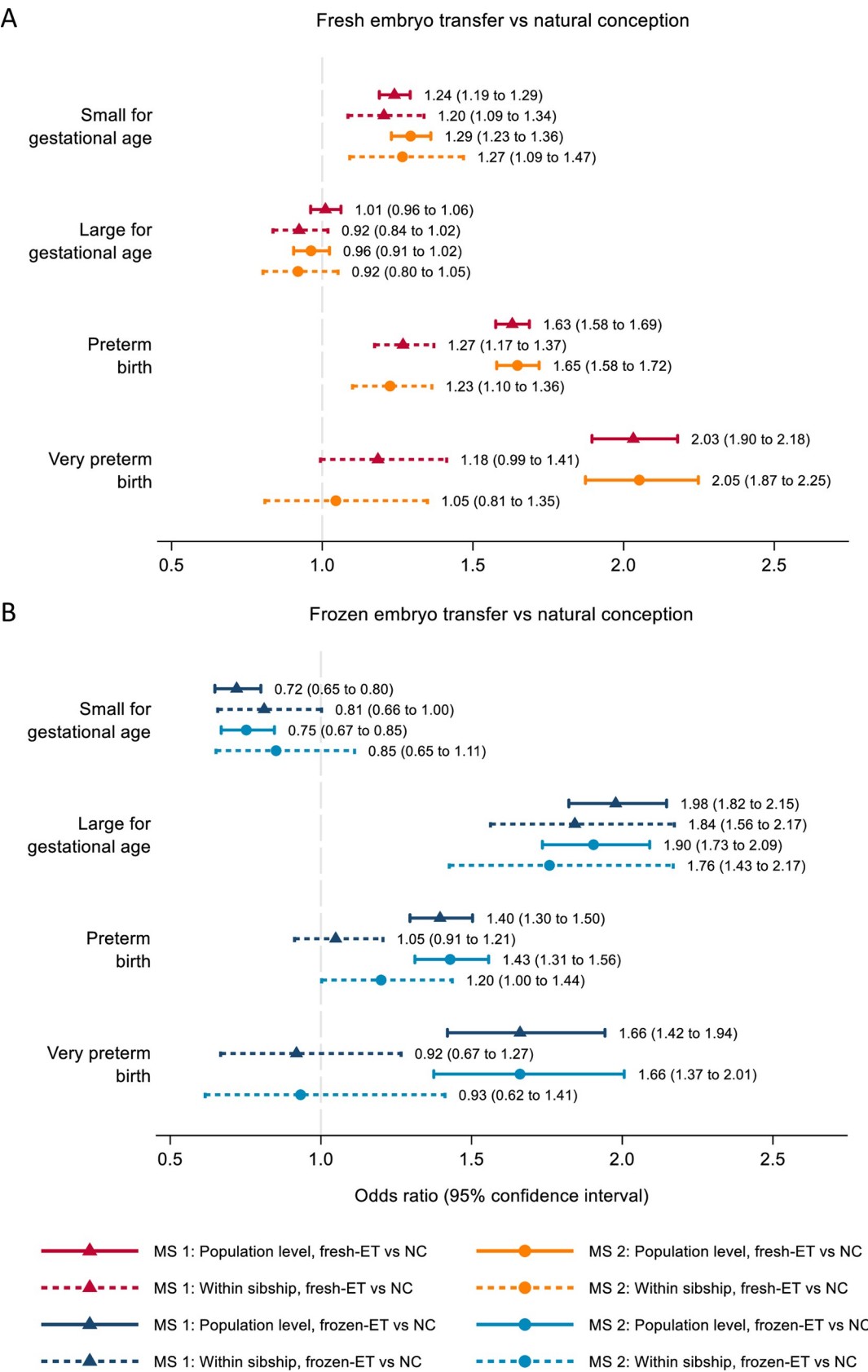

**Fig 2. Adverse perinatal outcomes according to conception method: Population estimates and offspring sibling comparison.** Odds ratios with 95% confidence intervals for fresh embryo transfer (fresh-ET) versus natural conception (NC) (A) and frozen embryo transfer (frozen-ET) versus natural conception (B). Main sample 1 (MS 1) estimates are adjusted for maternal age, parity, offspring birth year, and country (population level only) and minimize selection bias. Main sample 2 (MS 2) estimates are additionally adjusted for maternal body mass index, smoking status, and height (population level only) and minimize confounding. Fig 3A–3C shows the mean birthweights and risks of small and large for gestational age for a given birth order among sibling pairs with different combinations of conception methods. Overall, mean birthweight and risk of large for gestational age were greater, and risk of small for gestational age lower, in second-born compared to first-born siblings in all groups. Infants born after fresh-ET had the lowest birthweights and highest risk of small for gestational age for their birth order, while children born after frozen-ET had the highest birthweights and highest risk of large for gestational age, regardless of the conception method of their respective siblings.

results after more confounder adjustments, including for maternal BMI and smoking, both strongly associated with adverse perinatal outcomes [31–33] and largely missing from similar previous studies [20–23]. As reporting of BMI was introduced and improved during the study period, the children in main sample 2 comprise a more recent study population, reflecting contemporary treatment practice. Our sample was considerably larger than any previous population and included over 7-fold the number of discordant siblings compared with the 2 previous sibling studies directly comparing fresh-ET to frozen-ET.

A major strength was the comparison of siblings born after different conception methods. While the results from conventional population analyses are prone to residual confounding from unmeasured maternal and family characteristics, such as maternal health and family socioeconomic position, we expect the sibship analysis to account for many of these confounders as they are highly likely to be the same or very similar for siblings. Even if some characteristics may change between a woman's pregnancies, they are more likely to be similar within women than between women, and, therefore, the within-sibship analyses provide extra control for these characteristics. In addition, the large sample size supported analyses comparing the risk of outcomes according to order of conception methods used, as well as several sensitivity analyses that accounted for possible differences between maternal and full siblings, greater differences in maternal or family characteristics between siblings born more than 3 years apart, and the use of single embryo transfer and blastocyst culture. We found similar results for birthweight outcomes in all our approaches and populations, strengthening the evidence that type of ART treatment influences birthweight outcomes. For duration of pregnancy, results were also broadly consistent, and collectively they support that both ART treatments increase the risk of preterm birth, without clearly influencing risk of very preterm birth.

All birth institutions and ART clinics in the study countries adhere to a policy of mandatory reporting, ensuring valid and exhaustive data collection. Even so, women who receive cross-border reproductive care are likely to be misclassified as having natural conceptions in our study because they do not appear in the national ART registries. These will be a small group compared to the large group of correctly classified naturally conceiving women [34,35], and are therefore unlikely to substantially bias the results. Smoking was self-reported and could only be harmonized across all countries as a dichotomous variable. Further, smoking is commonly underreported among pregnant women [36] and is a source of residual confounding that we expect to be considerably worse in the population than within the sibship analyses. Estimation of gestational age in comparisons of natural and ART conception is challenging because fetuses from both fresh-ET and frozen-ET may have a greater estimated fetal size by ultrasound in both the first and second trimester compared to naturally conceived fetuses [37]. ART-conceived pregnancies may therefore be expected to have a higher gestational age when estimated from ultrasound measurements than from transfer date. Whether clinicians took this into consideration when determining gestational age is not known in our data. Our data from Denmark and Norway allowed comparison of the 2 methods of determining gestational

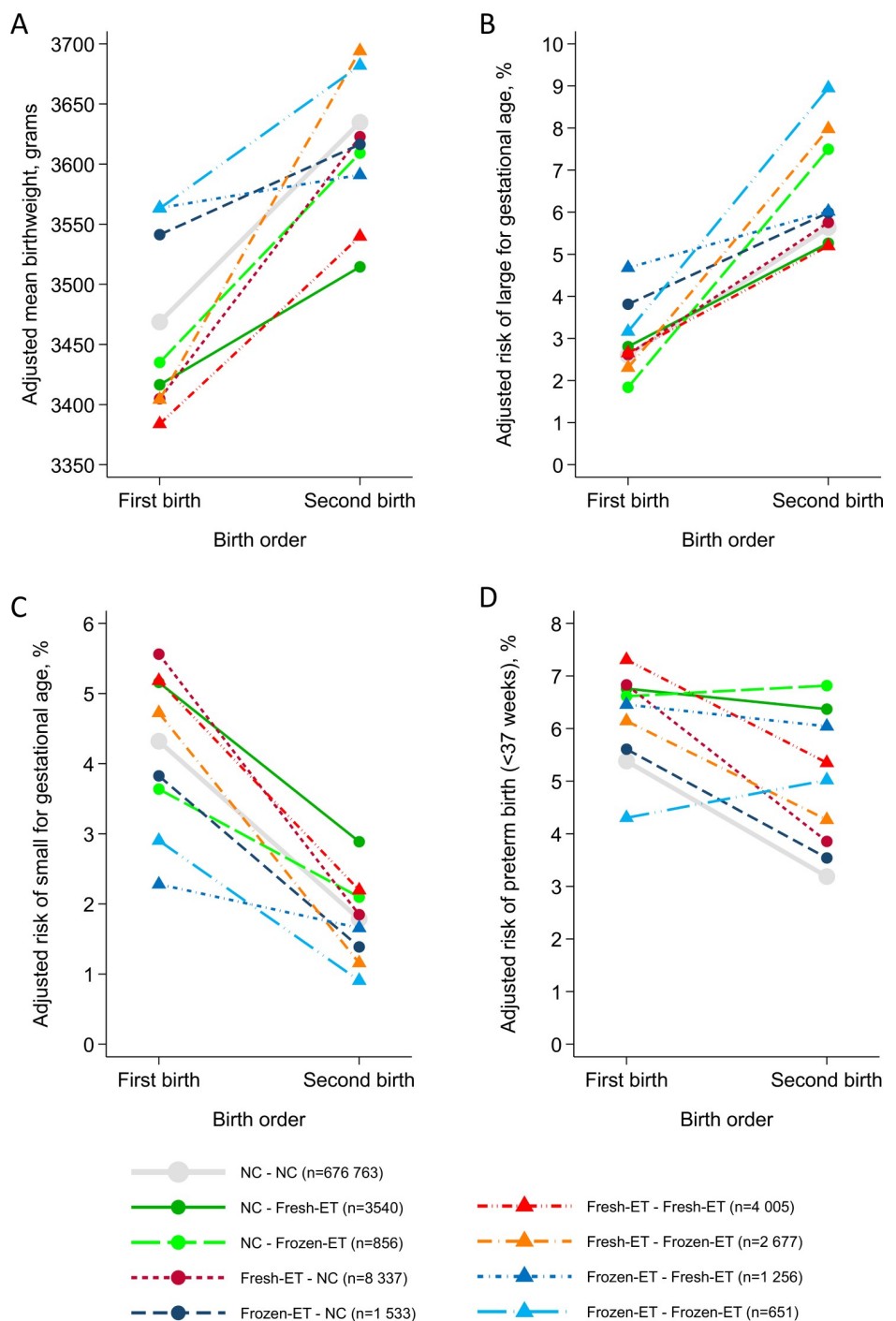

**Fig 3. Perinatal outcomes in consecutive offspring sibling pairs according to birth order and conception methods.** Means and absolute risks are estimated in main sample 2, using random effects logistic models with post-estimation commands. Adjusted for maternal age, offspring birth year, country, maternal body mass index, smoking status, and height. Fresh-ET, fresh embryo transfer; Frozen-ET, frozen embryo transfer; NC, natural conception.

age and indicated very similar distributions. As a result, we decided to use the ultrasound measurements since these are used in clinical management. Another limitation is the lack of information on embryo culture medium, which has been shown to affect perinatal outcomes [38].

Even if culture medium differed between clinics and over time, this should not specifically differ between fresh-ET and frozen-ET, which makes it less likely that our results are confounded by culture medium.

We pooled data from Denmark, Sweden, and Norway, and assume that results are consistent across these 3 countries. We believe this is a reasonable assumption given that they are all high-income Nordic countries with accessible and affordable healthcare systems that provide similar fertility treatment and perinatal care [5]. However, this may limit the generalizability to other populations.

## Comparison with other studies

We are not aware of previous studies of this size where perinatal outcomes of children born after fresh-ET or frozen-ET conception are compared to those of naturally conceived siblings as well as being explored using conventional population analyses. Previous studies that used within-sibship analysis to explore the associations of ART treatments with birth size and pregnancy duration had different designs and varied considerably in the covariates included [18–23]. While we defined birthweight for gestational age based on intrauterine growth curves [26], previous studies used observed birthweight and different criteria for large and small for gestational age [18–22]. As preterm birth often results from pregnancy complications that can affect fetal growth [39], the observed birthweights in preterm deliveries are not representative of the normal fetal weight distribution for healthy pregnancies at a given gestational age. Despite these differences, the results are broadly consistent (see Table M in S1 Text for a summary of study characteristics and estimates). In a Danish study of singletons born in 1994–2006, the results were largely consistent with ours, with lower mean birthweight and higher odds of low birthweight ($<$2,500 g) and preterm birth for children born after fresh-ET compared to their naturally conceived siblings (3,879 pairs), but no difference for very preterm birth [23]. Furthermore, infants born after frozen-ET had higher mean birthweights and lower odds of preterm birth than their siblings born after fresh-ET (358 pairs). A US study of ART-conceived singletons born in 2004–2013 compared fresh-ET to frozen-ET within sibships (3,681 discordant pairs) and found that siblings conceived with frozen-ET had greater odds of large for gestational age than those conceived with fresh-ET, but similar duration of pregnancy [22].

In 4 previous studies comparing ART-conceived infants to their naturally conceived siblings, conclusions were conflicting [18–20,23]. However, directions of associations were similar across these studies and magnitudes similar in several. Different conclusions may therefore reflect their different sample sizes and associated variation in power to detect statistical evidence. A Norwegian study of 2,204 sibling pairs born in 1988–2006 [18], a Dutch study of 1,813 sibling pairs born in 1999–2007 [20], and a Finnish study of 578 sibling groups born in 1995–2000 [19] showed no strong statistical support for associations, but associations in all 3 studies were in the direction of lower birthweight and gestational age in infants conceived after ART (either fresh-ET or frozen-ET) compared to their naturally conceived siblings. Although these studies did not provide separate estimates for fresh-ET and frozen-ET, their results would be expected to mainly reflect fresh-ET, which was by far the more common treatment during the study periods. A US study including 6,458 discordant sibling pairs born in 2000–2010 showed lower birthweight and gestational age after ART compared to natural conception, with stronger statistical support than the other studies, but, as with those studies, did not separate fresh-ET and frozen-ET [21].

We could not distinguish "freeze-all" cycles, a strategy to prevent ovarian hyperstimulation syndrome [40], from frozen-ET after an initial fresh transfer. However, in a recent study by Smith et al. [41], perinatal outcomes after a planned freeze-all cycle were similar to those after

frozen-ET in the conventional setting. This is in accordance with our study, where order of conception method was not associated with the perinatal outcomes.

In addition to the small number of previous within-sibship analyses described above, we also find some consistency with previous conventional observational studies, in which fresh-ET was associated with low birthweight and high risk of preterm and very preterm birth [8,9]. Frozen-ET, on the other hand, has been consistently associated with high birthweights, and some reports also indicate a lower risk of preterm birth compared to fresh-ET [8,9].

## Implications of findings and conclusion

We provide important evidence on the likely impact of fresh-ET compared with natural conception and of frozen-ET compared with natural conception. Infants born large for gestational age have a higher risk of delivery complications, and being born small for gestational age, large for gestational age, and preterm are all associated with increased perinatal morbidity and mortality [42,43]. They are also associated with long-term adverse outcomes [43–45]. Small for gestational age and preterm birth are associated with increased risk of cardiovascular diseases, mental health disorders, and social difficulties [44,45], and large for gestational age is associated with a higher risk of obesity and obesity-related adverse outcomes [45]. To ensure informed decision-making for infertile couples, and couples who are considering postponing childbearing, knowledge about adverse perinatal outcomes and their potential long-term consequences should be balanced against couples' desire to have a family at a time that suits them. Future studies should address whether close antenatal monitoring beyond present guidelines may improve perinatal outcomes in ART-conceived pregnancies.

The increased risk of large for gestational age and higher mean birthweight seen after frozen-ET has potential implications for the recent increase in freeze-all approaches [46], in particular when evidence from a recent large cohort study and a randomized trial suggests no benefit from freezing all embryos compared with an initial fresh transfer with respect to the cumulative live birth rate [41,47]. It has been suggested that the freeze-all approach should be limited to couples with a clinical indication, such as where the risk of maternal ovarian hyperstimulation syndrome is high [46,47]. Our findings add to the debate about the role of freeze-all strategies, by providing indirect evidence that it may not reduce adverse perinatal outcomes compared to fresh-ET followed by frozen-ET.

In this study we found that frozen-ET was associated with increased birthweight and risk of large for gestational age, whereas fresh-ET was associated with the opposite. Furthermore, sibship comparisons indicated that both fresh-ET and frozen-ET were associated with increased risk of preterm birth but not with risk of very preterm birth, despite strong associations in conventional population analyses. These findings should contribute to the ongoing discussions on the role of emerging ART approaches, such as the freeze-all approach, and to informed decision-making by couples and their healthcare providers. They should prompt studies to identify possible mechanisms and preventive measures to improve perinatal health in ART-conceived children.

## Supporting information

**S1 STROBE Checklist. Strengthening the Reporting of Observational Studies in Epidemiology (STROBE) guideline.**
(DOCX)

**S1 Study Protocol. The prospective analysis plan.**
(PDF)

**S1 Text.** Tables of main (Tables A and B) and sensitivity analyses (Tables C–L), and summary of previous sibship studies (Table M).
(DOCX)

## Acknowledgments

We thank all staff in the ART clinics and labor departments in the 3 contributing countries, for taking time to complete the ART registration forms and birth notifications in their busy working day. The details and completeness provide a solid foundation for our study.

## Transparency

The first and last author (KWJ and SO) have had full access to the data and affirm that this paper is an honest, transparent, and accurate account of the study and that no important aspects of the study have been omitted

## Dissemination to participants and related patient and public communities

It will not be possible to send the results to the study participants, but we plan to disseminate the findings to the public through media channels, our institutions' websites, and the CoNAR-TaS website (http://www.conartas.com).

## Author Contributions

**Conceptualization:** Liv Bente Romundstad, Christina Bergh, Mika Gissler, Anna-Karina A. Henningsen, Ulla-Britt Wennerholm, Aila Tiitinen, Anja Pinborg, Signe Opdahl.

**Data curation:** Liv Bente Romundstad, Christina Bergh, Mika Gissler, Anna-Karina A. Henningsen, Ulla-Britt Wennerholm, Aila Tiitinen, Anja Pinborg, Signe Opdahl.

**Formal analysis:** Kjersti Westvik-Johari, Signe Opdahl.

**Funding acquisition:** Liv Bente Romundstad, Christina Bergh, Mika Gissler, Ulla-Britt Wennerholm, Aila Tiitinen, Anja Pinborg, Signe Opdahl.

**Investigation:** Kjersti Westvik-Johari, Liv Bente Romundstad, Deborah A. Lawlor, Christina Bergh, Mika Gissler, Anna-Karina A. Henningsen, Ulla-Britt Wennerholm, Aila Tiitinen, Anja Pinborg, Signe Opdahl.

**Methodology:** Kjersti Westvik-Johari, Liv Bente Romundstad, Deborah A. Lawlor, Siri E. Håberg, Signe Opdahl.

**Project administration:** Anja Pinborg, Signe Opdahl.

**Resources:** Kjersti Westvik-Johari, Signe Opdahl.

**Software:** Anja Pinborg.

**Supervision:** Liv Bente Romundstad, Siri E. Håberg, Signe Opdahl.

**Validation:** Kjersti Westvik-Johari, Liv Bente Romundstad, Deborah A. Lawlor, Christina Bergh, Mika Gissler, Anna-Karina A. Henningsen, Siri E. Håberg, Aila Tiitinen, Anja Pinborg, Signe Opdahl.

**Visualization:** Kjersti Westvik-Johari, Signe Opdahl.

**Writing – original draft:** Kjersti Westvik-Johari.

**Writing – review & editing:** Kjersti Westvik-Johari, Liv Bente Romundstad, Deborah A. Lawlor, Christina Bergh, Mika Gissler, Anna-Karina A. Henningsen, Siri E. Håberg, Ulla-Britt Wennerholm, Aila Tiitinen, Anja Pinborg, Signe Opdahl.

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
