## [Editor Report · Decision Letter 0]

12 Nov 2020

Dear Dr Westvik-Johari, 

Thank you for submitting your manuscript entitled "Perinatal health after fresh and frozen embryo transfer in assisted reproduction - separating parental and treatment contributions. A cohort study with within sib-ship analysis" for consideration by PLOS Medicine.

Your manuscript has now been evaluated by the PLOS Medicine editorial staff and I am writing to let you know that we would like to send your submission out for external assessment.

Kind regards,

Richard Turner, PhD

Senior editor, PLOS Medicine

rturner@plos.org

---

## [Decision Letter · Decision Letter 1]

15 Dec 2020

Dear Dr. Westvik-Johari,

Thank you very much for submitting your manuscript "Perinatal health after fresh and frozen embryo transfer in assisted reproduction - separating parental and treatment contributions. A cohort study with within sib-ship analysis" (PMEDICINE-D-20-05488R1) for consideration at PLOS Medicine. 

Your paper was evaluated by the editors and sent to independent reviewers, including a statistical reviewer. The reviews are appended at the bottom of this email and any accompanying reviewer attachments can be seen via the link below:

[LINK]

In light of these reviews, we will not be able to accept the manuscript for publication in the journal in its current form, but we would like to invite you to submit a revised version that addresses the reviewers' and editors' comments fully. You will appreciate that we cannot make a decision about publication until we have seen the revised manuscript and your response, and we expect to seek re-review by one or more of the reviewers. 

We hope to receive your revised manuscript by Jan 11 2021 11:59PM. Please email us (plosmedicine@plos.org) if you have any questions or concerns.

Please let me know if you have any questions. Otherwise, we look forward to receiving your revised manuscript in due course. 

Sincerely,

Richard Turner, PhD

rturner@plos.org

To your data statement, please add (non-author) contact information for the study administrators or Statistics Denmark. 

Considering the observational study design, please avoid language that implies causality (e.g., "effects" early in your abstract). 

Please adapt your abstract to a three-part structure - you may find it helpful to consult one or two recent research papers published in PLOS Medicine to get a sense of the preferred style. 

The final sentence of the "Methods and findings" subsection of your abstract should begin "Study limitations include ..." or similar and quote 2-3 of the study's main limitations. 

Please adapt the text to "We found ... had ..." towards the end of your abstract. 

After the abstract, please add an "Author summary" section in non-identical prose, aiming to use the active voice in 1-2 points (e.g., "We found that ..."). 

Early in the methods section of your main text, please state whether the study had a protocol or prespecified analysis plan, and if so attach the relevant document(s) as a supplementary file(s), referred to in the text. Please highlight analyses that were not prespecified. 

Throughout the text, please adapt reference call-outs to the following style: "... socioeconomic position [16,17]." (i.e., with no spaces within the square brackets). 

Throughout the paper, please quote p values alongside 95% CI where available. 

Please ensure that all reference citations meet journal style. Six author names should be listed rather than 3, followed by "et al."; italics should be converted to plain text. 

Please add a completed checklist for the most appropriate reporting guideline - which may be STROBE or RECORD - as a supplementary document. Please refer to this in the methods section of your main text (e.g., "See S1_RECORD_Checklist"). In the checklist, please refer to individual items by section (e.g., "Methods") and paragraph number rather than by line or page numbers, as the latter generally change in the event of publication. 

Comments from the reviewers:

*** Reviewer #1: 

"Perinatal health after fresh and frozen embryo transfer in assisted reproduction - separating parental and treatment contributions. A cohort study with within sib-ship analysis" examines the correlations between fresh embryo transfer (fresh-ET) and frozen embryo transfer (frozen-ET), to various outcomes such as birthweight and preterm birth. Over 33,000 discordant sibling pairs from three countries were included, with two main analyses performed to minimize selection bias and minimize confounding, respectively. Various sensitivity analyses were also performed to study the effect of additional potential confounders. It was consistently found that infants conceived from fresh-ET tend to have lower birthweight/being small for gestational age, while those conceived from frozen-ET tend to have higher birthweight/being large for gestation age. Both fresh-ET and frozen-ET had increased risk of preterm birth. These findings largely concur with prior literature, which however did not generally account for confoundings as thoroughly.

The analyses appear convincing and robust on the whole, particularly as the sibship constraint helps to mitigate a large number of potential confounders, due to sharing the same parent(s).

1. It is stated that small for gestational age (SGA) was defined as birthweights less than 2 std dev, and large for gestational age (LGA) as birthweights greater than 2 std dev, where one standard deviation was set as 11% of the expected birthweight. However, it appears that the most common definition would be a weight below 10th percentile for SGA (and conversely above 90th percentile for LGA). Was there any reason as to why the 2 std dev definition was chosen instead (e.g. concurs with previous studies), and how far does it differ from the bottom/top 10th percentile definition, in practice?

2. For Sensitivity Analysis 2 in Figure 1, n=1,972,012 for naturally conceived infants, but n=1,982,791 in Supplementary Tables 5 & 6. This might be addressed.

Minor issues:

3. In the abstract, "95% 70 to 94" might be "95% CI".

4. The abbreviations SGA, LGA, PTB and VPTB (probably relating to gestational age and preterm births) in Figure 2, might be defined at some prior point in the paper (possibly within the figure caption definitions, as done for natural conception for example)

*** Reviewer #2: 

General comments

This is a retrospective cohort study within sub-ship analysis examining the perinatal outcome of babies born after fresh-ET and frozen-ET versus natural conception

Due to the well-known quality of ART-registration in Denmark, Norway and Sweden the authors can provide very interesting information on perinatal outcoma after assisted reproduction. 

The number of 33 056 offspring siblings with the same mother with at least two different conception methods is very important. Above this not only fresh-ET and frozen-ET offspring are compared, buth both groups are als compared with natural conception offspring in a huge number of patients.

I believe that this retrospective analysis, with even data of the paternal identity in 98 % of cases, deliveres results uncomparable with previous reports, not only due to the amount of cases, but also due to the methods used to prevent as much as possible confounding factors including socioeconomic status.

This paper can be regarded as a major contribution in our understanding how assisted reproduction can influence perinatal outcome results when compared to natural conception.

Specific comments

Only two comments:

Why did the authors looked at mean birthweight and small for getational age. In the literature <2.5kg and < 1.5 kg is mostly used.

There is no mention of the influence of different culture media on birthweight in the discussion. I assume that this parameter could not be examined.

*** Reviewer #3: 

This is the largest study investigating perinatal outcomes following fresh and frozen embryo transfers (ETs) compared with natural conception (NC) within siblings. The sample size is extremely larger (33,056 sibling pairs) than other sibling analysis studies. Further, the study stratified the analysis according to fresh/frozen status, which have not been done in most of previous studies. Though significant associations between ART and perinatal outcomes were attenuated and did not become significant within sib-ship analysis in many previous sibling studies, , this study found that significant association between fresh ET and LBW, SGA, and between frozen ET and higher birthweight and LGA. which was similar in population analysis. Only inconsistent result between population analysis and sibling analysis was very preterm birth. 

The author took a strict strategy for sample selection to manage selection bias and confounding. Further, they repeated several sensitivity analyses to test the robustness of the associations. I have only several minor points that should be addressed before publication.  

1. Discussion (P 18): Though the author mentioned about other studies for comparison, I do not think those studies are broadly consistent results. One recent study from the Lancet (Goisis et al.) demonstrated significant associations were very highly attenuated after statistical adjustment. Another study from the Netherlands (Seggers et al.) using 1813  sibling pairs demonstrated null association. Although the author mentioned the direction of the association was similar, non-significant associations did not have any directions. Further, conclusions of those previous studies were opposite to the current study. Thus, I recommend to describe more in detail about the discrepancy for study results, potential reasons and future implication on it.

***

[LINK]

---

## [Editor Report · Decision Letter 2]

23 Jan 2021

Dear Dr. Westvik-Johari,

Thank you very much for re-submitting your manuscript "Perinatal health after fresh and frozen embryo transfer in assisted reproduction - separating parental and treatment contributions. A cohort study with within sib-ship analysis" (PMEDICINE-D-20-05488R2) for consideration at PLOS Medicine.

I have discussed the paper with editorial colleagues and our academic editor and I am pleased to tell you that, provided the remaining editorial and production issues are fully dealt with, we expect to be able to accept the paper for publication in the journal.

[LINK]

In revising the manuscript for further consideration here, please ensure you address the specific points made by the editors. In your rebuttal letter you should indicate your response to the reviewers' and editors' comments and the changes you have made in the manuscript. Please submit a clean version of the paper as the main article file. A version with changes marked must also be uploaded as a marked up manuscript file.

Please let me know if you have any questions. Otherwise, we look forward to receiving the revised manuscript shortly.   

Sincerely,

Richard Turner, PhD

rturner@plos.org

Requests from Editors:

We suggest adapting the title to: "Separating parental and treatment contributions to perinatal health after fresh and frozen embryo transfer in assisted reproduction: a cohort study with within-sibship analysis".

Please quantify these results in the abstract with ORs, 95% CIs and p values: “For very preterm birth, population analyses showed increased odds for both fresh- and frozen- ET compared with natural conception, but results were notably attenuated within siblings.”

Please ensure that numbers are quoted consistently throughout the paper, noting "4,414,705" in table 1. 

In table 3, please make that "CI - confidence interval".

The first paragraph of the discussion should summarize the study - therefore, please use the past tense to summarize the findings (e.g., "we found that children born after ART were at higher risk ..."). 

Please correct "ashorter" in the first paragraph of the Discussion section. 

In the final paragraph of the Discussion section, please begin the first sentence with "In this study, we found that frozen-ET was associated with ..." or similar, and adapt the subsequent text to the same tense. 

Please convert "within siblings" to "between siblings" throughout the paper, where appropriate. 

Throughout the text, please ensure that reference call-outs precede punctuation and remove spaces from within the square brackets (e.g., "... morbidity and mortality [48,49].").

Please move the information on study ethics approval from the end of the main text to the methods section. 

Please remove the information on funding and competing interests from the end of the main text. Upon publication this information will appear in the article metadata via entries in the submission form. 

Please ensure that all references meet journal format. For example, all italic and boldface text should be converted to plain text. Six author names should be listed where appropriate, followed by "et al.". All instances of "p." preceding page numbers should be removed. Journal names should be abbreviated consistently. 

For reference 16, for example, author names should be listed as : "Lawlor D, Tilling K, Davey Smith G."

Please rename the attached protocol document to the label by which it is referred to in the text.

***

---

## [Editor Report · Decision Letter 3]

3 Jun 2021

Dear Dr Westvik-Johari, 

On behalf of my colleagues and the Academic Editor, Dr Smith, I am pleased to inform you that we have agreed to publish your manuscript "Separating parental and treatment contributions to perinatal health after fresh and frozen embryo transfer in assisted reproduction: a cohort study with within-sibship analysis" (PMEDICINE-D-20-05488R3) in PLOS Medicine. We do apologize for the delay in sending you this decision. 

Prior to final acceptance, please: remove the information about data provision from the end of the main text (this will appear in the article metadata via entries in the submission form; and remove the competing interest information from reference 42. We suggest "within sibship" and "between siblings", for example, throughout, but leave this to you. 

PRESS

Sincerely, 

Richard Turner, PhD 

rturner@plos.org